

# Growth and fatty acid characterization of microalgae isolated from municipal waste-treatment systems and the potential role of algal-associated bacteria in feedstock production

Kevin Stemmler, Rebecca Massimi and Andrea E. Kirkwood

Faculty of Science, University of Ontario Institute of Technology, Oshawa, Ontario, Canada

## ABSTRACT

Much research has focused on growing microalgae for biofuel feedstock, yet there remain concerns about the feasibility of freshwater feedstock systems. To reduce cost and improve environmental sustainability, an ideal microalgal feedstock system would be fed by municipal, agricultural or industrial wastewater as a main source of water and nutrients. Nonetheless, the microalgae must also be tolerant of fluctuating wastewater quality, while still producing adequate biomass and lipid yields. To address this problem, our study focused on isolating and characterizing microalgal strains from three municipal wastewater treatment systems (two activated sludge and one aerated-stabilization basin systems) for their potential use in biofuel feedstock production. Most of the 19 isolates from wastewater grew faster than two culture collection strains under mixotrophic conditions, particularly with glucose. The fastest growing wastewater strains included the genera *Chlorella* and *Dictyochloris*. The fastest growing microalgal strains were not necessarily the best lipid producers. Under photoautotrophic and mixotrophic growth conditions, single strains of *Chlorella* and *Scenedesmus* each produced the highest lipid yields, including those most relevant to biodiesel production. A comparison of axenic and non-axenic versions of wastewater strains showed a notable effect of commensal bacteria on fatty acid composition. Strains grown with bacteria tended to produce relatively equal proportions of saturated and unsaturated fatty acids, which is an ideal lipid blend for biodiesel production. These results not only show the potential for using microalgae isolated from wastewater for growth in wastewater-fed feedstock systems, but also the important role that commensal bacteria may have in impacting the fatty acid profiles of microalgal feedstock.

Corresponding author
Andrea E. Kirkwood,
andrea.kirkwood@uoit.ca

## INTRODUCTION

World energy consumption of petroleum is currently estimated to be 4.4 billion tonnes per year to fuel electricity, automobiles, and industrial processes (*Roddy, 2013*). To meet the growing global energy demand, nations worldwide have considered renewable sources

of fuel to offset the dependence on non-renewable sources like fossil fuels. Algal biofuels are a promising source of renewable energy since production is non-seasonal, and yields are not limited to one or two harvests per year (*Da Silva et al., 2009*). When compared to other biofuel feedstocks like corn or switchgrass, algal biofuel crops possess 6–12 times greater yearly energy production (*Sandefur, Matlock & Costello, 2011*). An additional benefit of algal feedstock relates to the variety of possible conversion processes such as direct combustion, pyrolysis, or chemical conversion. Algal biomass can be converted to fuels like hydrogen, oil, or even raw electricity (*Brennan & Owende, 2010*; *Tsukahara & Sawayama, 2005*).

Research in algal biofuels has been primarily directed towards improving algal biomass yield or increasing lipid content. Researchers have determined optimal temperatures, light conditions, and nutrients to obtain high growth rates for many strains (*Chinnasamy et al., 2010*; *Price, Yin & Harrison, 1998*). These parameters are not the same for each strain of microalgae, and as a result, they need to be optimized individually. One challenging aspect of growing microalgae compared to terrestrial plant crops is the sizeable amount of water required to sustain yields. To grow algal biofuel-feedstock at large-scale cultivation would require approximately 1.5 million liters of water per hectare (*Cho et al., 2007*). If grown in open ponds, the evaporation loss would be 7–11 million liters of water per hectare per year (*Cho et al., 2007*). This requirement is juxtaposed against the prediction that 66% of the world's population will be residing in regions without access to drinking water by 2025 (*Mehanna et al., 2010*). Additionally, *Sandstrom (1995)* showed that a 15% reduction in precipitation due to climate change may result in a 40–50% reduction in aquifer water recharge. As such, a major stumbling block for the economic and environmental feasibility of algal-based biofuel production is the potentially unsustainable requirement for freshwater.

In recent years, research has started to focus on exploiting wastewater rather than freshwater as a growth medium for algal feedstock production. Wastewater, particularly municipal and agricultural, typically contain elevated levels of essential nutrients such as nitrogen and phosphorous. For example, nitrogen and phosphorus concentrations in municipal wastewaters can range from 10–100 mg L$^{-1}$ and >1,000 mg L$^{-1}$ in agricultural wastewater (*De la Noue, Laliberte & Proulx, 1992*). Not only does wastewater serve as a good growth medium for microalgae, but conversely, algal growth in wastewater can serve as a tertiary treatment option (*De la Noue, Laliberte & Proulx, 1992*; *Lau, Tam & Wong, 1995*; *Woertz et al., 2009*). *Sandefur, Matlock & Costello (2011)* have shown that algal strains grown in treated wastewater were able to significantly decrease both phosphorus and nitrogen concentrations. When microalgae are grown in piggery wastewater, *An et al. (2003)* showed that the green alga *Botryococcus braunii* could remove 80% of the nitrate content. Another significant component of municipal wastewater is the organic load (*Rogers, 1996*; *Fadini, Jardim & Guimarães, 2004*). Sewage sourced from human populations is replete with biodegradable compounds such as sugars, amino acids and other breakdown products of digestion (*Painter & Viney, 1959*; *García, Hernández & Costa, 1991*). Although readily mineralized by bacteria during secondary treatment, these organic compounds can also be utilized by mixotrophic algae.

Even though wastewater has great potential as a growth medium for algal-feedstock production, a major caveat is the broad variation in wastewater quality and levels of toxic constituents. Most studies assessing algal growth in wastewater use isolates from lakes, rivers, or other naturally occurring water bodies (*Park et al., 2012*; *Zhou et al., 2011*). *Zhou et al. (2011)* reported that only a few strains from the genera *Chlorella* and *Scenedesmus* have been analyzed for their ability to grow in municipal wastewater. Some work (*Liu et al., 2011*) has been done to isolate and culture microalgae from wastewater treatment systems, but assessment of their potential use as a biofuel feedstock remains largely unknown. Also, most studies characterizing microalgae for biofuel production typically use axenic strains and sterile media (*Price, Yin & Harrison, 1998*; *Liu et al., 2011*; *Zhou et al., 2011*), which eliminates the potential role of bacteria in influencing algal growth and lipid production.

In order to move forward the prospects of large-scale cultivation of microalgae in wastewater, we set out to assess the potential of microalgae isolated from municipal wastewater treatment systems to serve as candidates for biofuel-feedstock production. The rationale for this is that microalgae isolated from municipal wastewater would have inherently higher tolerance to wastewater compared to lab strains or isolates from natural systems, as well as an increased capacity to grow mixotrophically to exploit organic compounds in wastewater. The first phase of our assessment is presented in this paper, which focuses on: (1) Isolating algal strains from municipal waste-treatment systems; (2) Characterizing the metabolic growth characteristics of wastewater isolates (i.e., photoautotrophy, mixotrophy, and heterotrophy), including their fatty acid composition; and (3) Assessing the effects of commensal bacteria on algal growth and lipid production compared to axenic strains.

## MATERIALS AND METHODS

### Strain collection and isolation

Wastewater samples were collected from three municipal wastewater treatment plants in southern Ontario, Canada including the towns of Port Perry and Whitby, and the city of Hamilton. All municipal wastewater treatment facilities were activated sludge, with the exception of Port Perry, which is a lagoon system of aerated stabilization basins. Wastewater from secondary (i.e., biological) treated effluent was collected in six sterile 1-L Nalgene$^{TM}$ bottles at mid- and final treatment stages. Wastewater samples were placed into a cooler and stored at 4 °C until enrichment cultures were initiated within 24-h of collection. Effluent (100-mL) was placed into sterile Erlenmeyer flasks in triplicate. Flasks were then placed on a cool-fluorescent light bank (12:12 dark/light cycle) to encourage algal growth in whole-effluent.

Once growth was confirmed either by naked eye or microscopic examination, 5-mL of the sample was transferred to a 250 mL Erlenmeyer flask containing either autoclaved BG11 (*Rippka et al., 1979*) or CHU10 (*Stein, 1973*) media. Two types of media were chosen in an attempt to isolate a broader diversity of isolates. After growth was established in media, 5-mL of culture was added to new medium and grown for two weeks. This process

**Table 1  List of microalgal strains.** List of microalgal strains assessed in this study including nineteen municipal wastewater treatment plant (WWTP) isolates and two Canadian Phycological Culture Collection (CPCC) reference strains.

| Taxonomic ID | Strain ID | Isolating medium | Treatment stage | Site of origin |
|---|---|---|---|---|
| *Botrydiopsis* | B1N | CHU10 | Mid-stage secondary treatment | Port Perry, Ontario WWTP |
| *Botrydiopsis* | B2H | BG11 | Final stage of secondary treatment | Hamilton, Ontario WWTP |
| *Chlorella kesslirii* | C1U | Unknown | CPCC Reference strain (CPCC 266) | Unknown (USA) |
| *Chlorella* | C3N | CHU10 | Mid-stage secondary treatment | Port Perry, Ontario WWTP |
| *Chlorella* | C4C | CHU10 | Mid-stage secondary treatment | Whitby, Ontario WWTP |
| *Chlorella* | C5C | CHU10 | Mid-stage secondary treatment | Whitby, Ontario WWTP |
| *Chlorella* | C6C | CHU10 | Mid-stage secondary treatment | Whitby, Ontario WWTP |
| *Dictyochloris* | D1N | CHU10 | Final stage of secondary treatment | Port Perry, Ontario WWTP |
| *Ellipsoidon* | E1H | BG11 | Final stage of secondary treatment | Hamilton, Ontario WWTP |
| *Ellipsoidon* | E2C | CHU10 | Mid-stage secondary treatment | Whitby, Ontario WWTP |
| *Ellipsoidon* | E3N | CHU10 | Final stage of secondary treatment | Port Perry, Ontario WWTP |
| *Microcystis* | M1H | BG11 | Mid-stage secondary treatment | Hamilton, Ontario WWTP |
| *Scenedesmus acutus* | S1B | Unknown | CPCC Reference strain CPCC 10 | Boucher Lake, ON |
| *Scenedesmus* | S2N | CHU10 | Final stage of secondary treatment | Port Perry, Ontario WWTP |
| *Scenedesmus* | S3N | CHU10 | Final stage of secondary treatment | Port Perry, Ontario WWTP |
| *Scenedesmus* | S4N | CHU10 | Mid-stage secondary treatment | Port Perry, Ontario WWTP |
| *Scenedesmus* | S5N | CHU10 | Final stage of secondary treatment | Port Perry, Ontario WWTP |
| *Scenedesmus* | S6H | BG11 | Mid-stage secondary treatment | Hamilton, Ontario WWTP |
| *Scenedesmus* | S7H | BG11 | Mid-stage secondary treatment | Hamilton, Ontario WWTP |
| *Scenedesmus* | S8C | CHU10 | Mid-stage secondary treatment | Whitby, Ontario WWTP |
| *Scenedesmus* | S9C | CHU10 | Mid-stage secondary treatment | Whitby, Ontario WWTP |

was repeated two successive times to ensure the compatibility of the algal culture under lab conditions (*Anderson, 2005*). Isolation from the algal consortium was undertaken by serial dilutions and transferring 100-μL of diluted culture to 1.5% agar for spread plating. Once single colonies were formed they were transferred with the use of a flame sterilized loop to a well in a 24-microwell plate containing 1-mL of either medium. When growth was present and unialgal cultures were confirmed microscopically, the cultures that were unialgal were transferred to fresh media. Morphologically unique strains were selected and maintained for further experimentation. Although several strains of the same genus and sampling location were isolated (Table 1), morphological variations witnessed under the microscope were used to discriminate between unique strains. Future 18S rRNA sequencing of strains will help to further delineate strain identification. However, past work (*Kirkwood et al., 2008*) has clearly shown that phylogenetic similarity does not necessarily reflect physiological similarity.

Before experimentation, each culture was re-evaluated microscopically to confirm unialgal status. Reference strains *Scenedesmus acutus* (CPCC 10) and *Chlorella kesslerii* (CPCC 266) were obtained from the Canadian Phycological Culture Collection (CPCC), Waterloo, Ontario, Canada to compare their growth characteristics to wastewater strains. A comparison between CHU10 and BG11 media revealed that after 5–10 days of growth, algal strains grown in CHU10 started to yellow while BG11 strains retained a healthy green

colour. As such, all subsequent cultures and growth experiments were conducted with BG11. The nitrogen concentration in BG11 was reduced by a factor of ten to bring it down to a level more closely aligned with wastewater concentrations.

### Experimental growth conditions

All algal growth experiments were conducted in an Algaetron$^{TM}$ environmental growth chamber (Algaetron Photon System Instruments, Czech Republic) with a built in shaker table. Growth parameters were set to 200 rpm and 150 µmol photons m$^{-2}$ s$^{-1}$. A temperature of 22 °C was used to match the ambient temperature in the lab to minimize any variations encountered when sampling. To assess the effect of growth condition (photoautotrophy, mixotrophy, heterotrophy) on growth rate and fatty acid accumulation, all three growth conditions were tested concurrently for each strain. For the photoautotrophic conditions, all organics were removed from the BG11 media (citric acid, ferric ammonium citrate, and sodium EDTA) and the iron was replaced with ferric chloride similar to the method described by *Kirkwood, Nalewajko & Fulthorpe (2003)*. The mixotrophic conditions utilized the organic-free medium from the photoautotrophic trials with the addition of either glucose (14 mM) or acetate (3 mM), similar to *Park et al. (2012)* and *Kirkwood, Nalewajko & Fulthorpe (2003)*. The heterotrophic growth trials utilized the same medium as the mixotrophic trials with the exception of being placed in a cardboard box to exclude illumination. Cultures for experimentation were grown in 50-mL Erlenmeyer flasks in triplicate for seven days. To standardize inocula among experiments and strains, mid-exponential phase inocula with similar cell densities were used. Standard curves of cell density for each algal strain were based on cell counts using a haemocytometer and optical density measured with a Genesys 10S UV-Vis Spectrophotometer (Thermo Scientific, Waltham, MA, USA). Growth rates for each algal strain were calculated from the slope of log-transformed 7-day growth curves ($d^{-1}$).

### Antibiotic treatment of algal isolates

In order to measure fatty acids derived only from the algal isolates, all strains were subjected to a series of antibiotic treatments to remove commensal bacteria. This was accomplished through the application of streptomycin and penicillin following the methods outlined by *Droop (1967)*. To verify axenic status after 24 h of incubation in the antibiotic media, a 1-mL aliquot of each dilution was transferred to 1.5% agar containing peptone via pour plate method. The plates were then placed on the light table and monitored over the course of 48 h for bacterial growth. If no growth was visible on the plates, microscopic examination was performed to verify axenic status. If a sampled failed to achieve axenic status the procedure was repeated. Strains were classified as non-axenic if either the antibiotics proved ineffective at eliminating the bacteria, or the antibiotics prevented algal growth. Parent, non-axenic cultures of all axenic strains were maintained for experiments comparing fatty acid profiles and neutral lipid concentrations.

### Fatty Acid Methyl Esters (FAME) analysis

Due to the extremely low biomass yields for all algal strains grown under heterotrophic conditions, only photoautotrophic and mixotrophic cultures were analysed for fatty acid

methyl esters (FAME). Both axenic and non-axenic forms of the same strain were included in FAME analysis. The lipid extraction procedure was based on the FAME synthesis described in *O'Fallon et al. (2007)*. In brief, 7-mL subsamples of microalgae with a cellular density $\geq 10^5$ cells·mL$^{-1}$ was collected at day seven of the experiment. These samples were centrifuged at 1,690 g force for 5 min. The supernatant was discarded and the samples were stored in $-20$ °C until freeze dried. Once freeze dried (typically 24–48 h) samples could be processed. The fatty acids derived via the lipid extraction method was determined by capillary Agilent J& W GC column (30 m·0.25 mm·0.25 μm) installed in a Varian 450 gas chromatograph. The initial oven temperature was 135 °C held for 4 min, followed by an increase to 250 °C at a rate of 4 °C min$^{-1}$, followed by a hold at this temperature for 10 min. The front injector was 250 °C and held for 20 min, the middle was 110 °C for 1 min, and the rear injector was held at 180 °C for 20 min. A mixture of helium and nitrogen was used as the carrier gas with a split ratio of 10:1 at a flow rate of 1-mL min$^{-1}$ for 3.5 min then shut off. The retention times of each sample were analyzed using an electron impact ion trap mass spectrometer. The standards used for analysis were carbon chain 8 to carbon chain 24 (C8–C24) along with a Bacterial Acid Methyl Ester (BAME) fatty acid mix. The BAME standard was used to filter out non-algal fatty acids. Fatty acids were identified through the use of MS workstation by comparing their retention times to known fatty acid methyl standards. Each chromatograph was then analyzed manually and any peaks that were above the background noise threshold were library searched to determine if the peak was an associated fatty acid. All standards were purchased from Sigma Aldrich (Oakville, ON, Canada).

## Quantification of neutral lipids

Similar to the FAME analyses, only photoautotrophic and mixotrophic cultures were analysed for neutral lipids. Both axenic and non-axenic forms of select strains were included in neutral lipids analysis. Neutral lipids in algal cells were determined using the standard Nile Red assay. A calibration curve for neutral lipids was based on the fluorescence response of varying triolein concentrations. Subsamples (1.5-mL) of algal culture from the growth experiments were stored at $-20$ °C prior to analysis. Due to the limited filter set available, we decided to use an excitation of 590 and emission of 640 nanometers based on the strong fluorescent response of the Nile Red dye at these wavelengths. Nile Red was dissolved in HPLC grade acetone at a concentration of 500 μg mL$^{-1}$, and was stored in complete darkness in a vial covered in aluminium foil to prevent photodegradation. In a black-opaque 96 microwell plate, 150-μL of algal sample was aliquoted into the wells in triplicate along with triplicate blanks. Following the addition of the algal samples to the wells, 90-μL of Nile Red was added to the treatment wells to dye the algal cells. The microwell plate was covered and incubated in darkness at room temperature for 10 min to allow dye penetration into the cells. After incubation, the plate was immediately analyzed with a Synergy HT multimode plate reader (BioTEK, Winooski, VT, USA).

## Data analysis

To determine statistically significant differences between growth condition treatments and strains, Sigma Plot 12 was used to run Analysis of Variance (ANOVA) on exponential

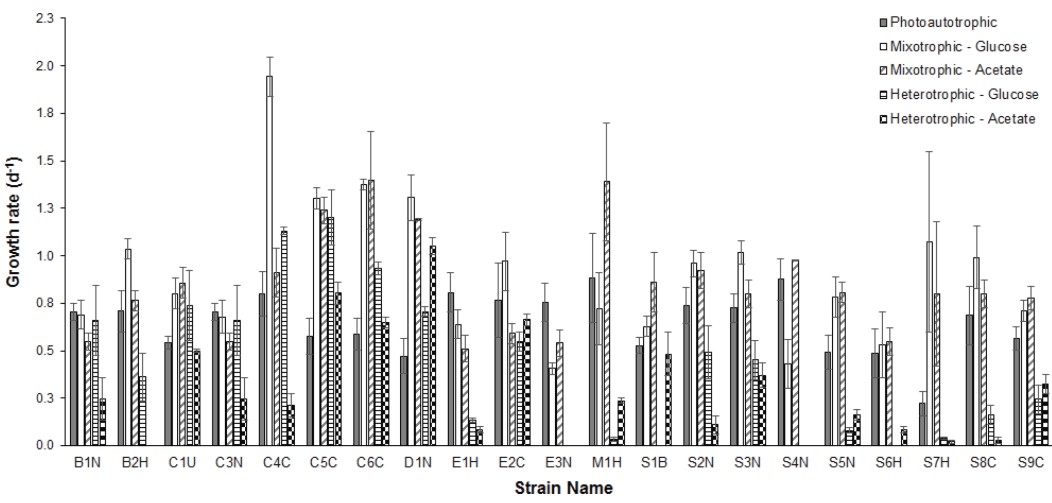

**Figure 1** **Comparison of growth rates under different conditions.** Comparison of mean ($n = 3$) exponential phase specific-growth rates between microalgal strains grown under photoautotrophic, mixotrophic, and heterotrophic conditions. Error bars reflect standard error of the mean.

growth rates and neutral lipid concentrations. A Shapiro–Wilk test was performed to determine if the data were normally distributed and an All Pairwise Multiple Comparison (Holm-Sidak method) was performed to detect significant differences between individual strains. The Paleontological Statistics (PAST) software package (v3) (*Hammer, Harper & Ryan, 2001*) was used for cluster analysis and Principal Component Analysis (PCA). Cluster analysis was based on the average growth rate of each of 19 strains under each of 5 growth conditions (photoautotrophy, mixotrophy-glucose, mixotrophy-acetate, heterotrophy-glucose, heterotrophy-acetate). PCA was based on the percent abundance of fatty acids identified in the FAME analysis, and least-squares linear regression was used to regress PCA axis scores against individual fatty acids.

## RESULTS

Microalgae were successfully isolated from all wastewater treatment plants. Table 1 provides a list of strains, media used to isolate each strain, and the original sampling location. Most of the 19 wastewater strains isolated were chlorophyte microalgae, dominated by *Scenedesmus* and *Chlorella* species. The only non-chlorophyte taxon isolated was the cyanobacterium *Microcystis*. With respect to the diversity of taxa isolated, Port Perry wastewater treatment plant provided strains from all isolated genera, followed by Hamilton and Whitby wastewater treatment plants. The application of antibiotic treatment rendered nine of the wastewater isolates axenic (Table 1).

Average growth rates ($n = 3$) for each strain grown under photoautotrophic, mixotrophic and heterotrophic conditions are presented in Fig. 1. When comparing the CPCC strains to their taxonomically-similar counterparts from wastewater systems, most *Chlorella* wastewater isolates (with the exception of C3N) grew significantly faster (ANOVA and Holm-Sidak, $p < 0.05$) than the CPCC reference strain C1U, particularly under mixotrophic growth conditions. In contrast, there were only a few *Scenedesmus* wastewater strains that

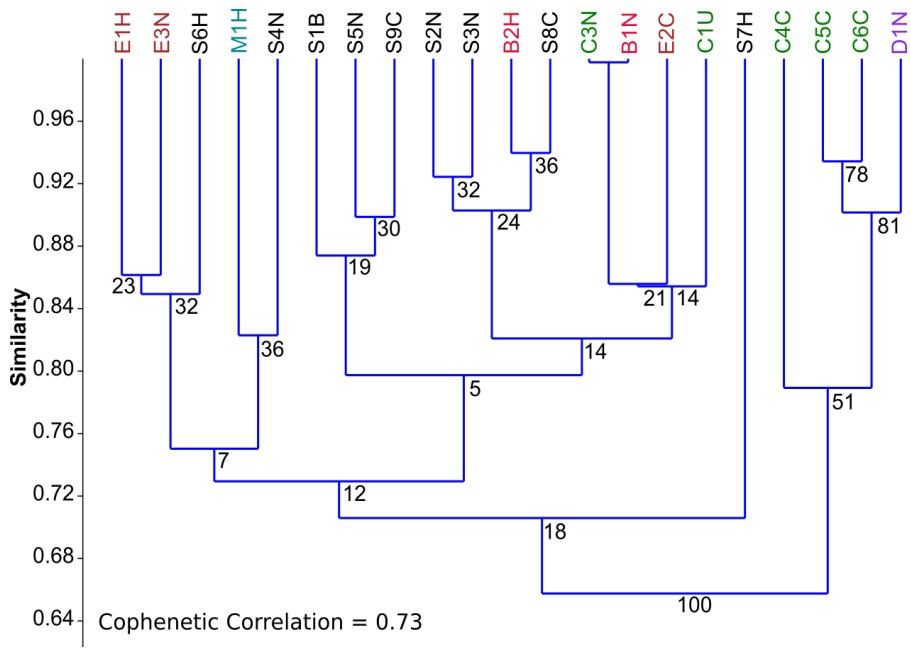

**Figure 2 Comparison of trophic profiles among microalgal strains.** Comparison of trophic profiles (photoautotrophic, mixotrophic, and heterotrophic) among strains using cluster analysis based on un-weighted pair-group average (UPGMA) using the Bray-Curtis Similarity Index showing Bootstrap analysis values (N=1,000) at each node.

grew significantly (ANOVA, $p < 0.05$) faster under mixotrophic conditions with glucose, compared to the CPCC reference strain S1B. In general, the *Scenedesmus* strains grew significantly slower (ANOVA, $p < 0.05$) than the *Chlorella* strains. The fastest growth rates for any strain were under mixotrophic conditions. One other comparably fast growing strain was D1N, which grew well heterotrophically on acetate. Although several strains could grow fairly well on glucose under heterotrophic conditions, including D1N and most of the *Chlorella* strains, most could not grow very well on either glucose or acetate under heterotrophic conditions.

To assess the trophic similarities of strains based on photoautotrophic, mixotrophic, and heterotrophic growth, cluster analysis was performed (Fig. 2). Although the cophenetic correlation coefficient (0.73) indicates that the hierarchical structure in the dendrogram represents the actual similarity moderately well, most dendrogram nodes had weak to moderate agreement based on bootstrap values (1,000 replicates). The dendrogram reveals some taxonomic clustering, but mostly a diversity of functional growth-types. The most physiologically similar strains belong to different taxonomic genera, such as *Chlorella* C3N and *Botrydiopsis* B1N. A few strains were less than 80% similar to all the other strains, including *Scenedesmus* S7H and *Chlorella* C4C.

Fatty acid profiles via FAME analysis were determined for each strain under photoautotrophic and mixotrophic conditions with glucose or acetate (Figs. 3A–3C). To improve clarity, strains were removed from the analysis if they had less than 1% fatty acids detected. Under photoautotrophic conditions (Fig. 3A), most non-axenic strains remained in the analysis with detectable fatty acids. Remaining axenic strains tended to
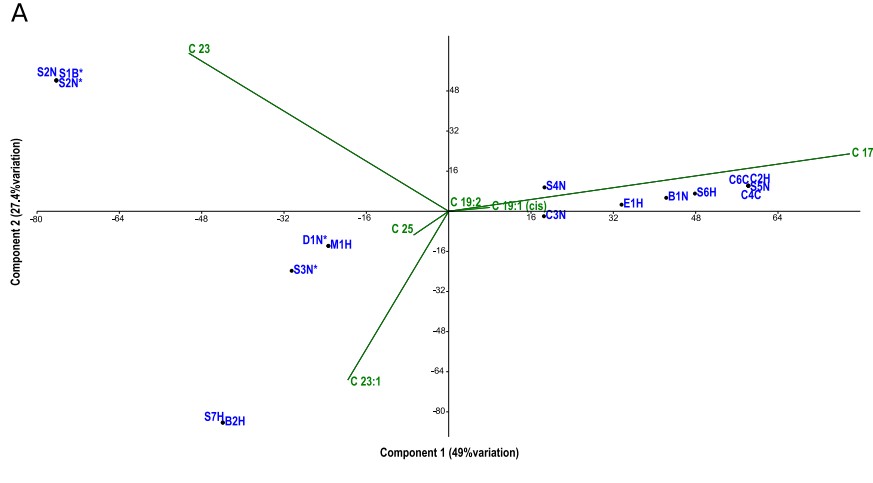

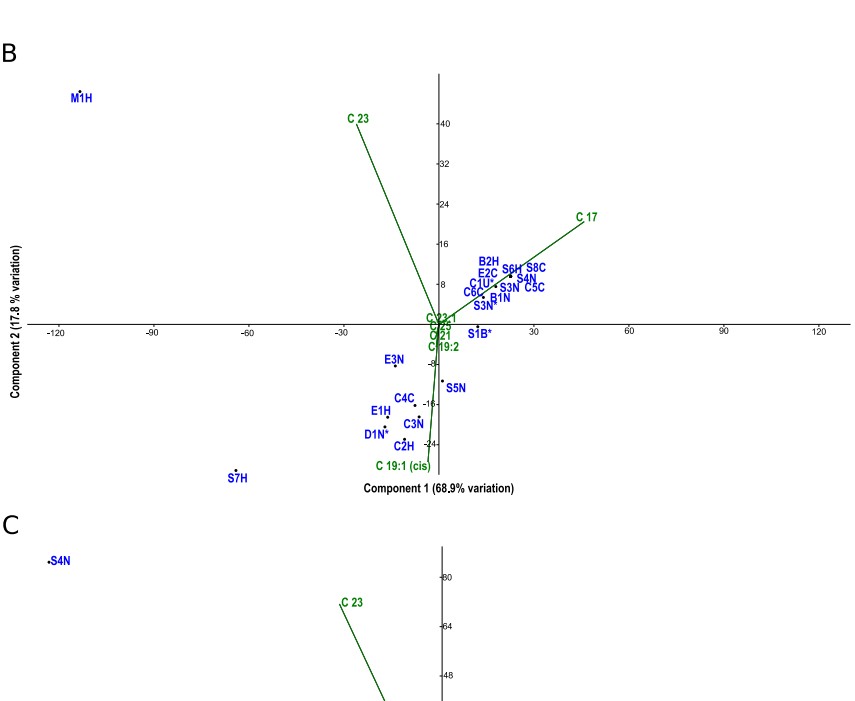

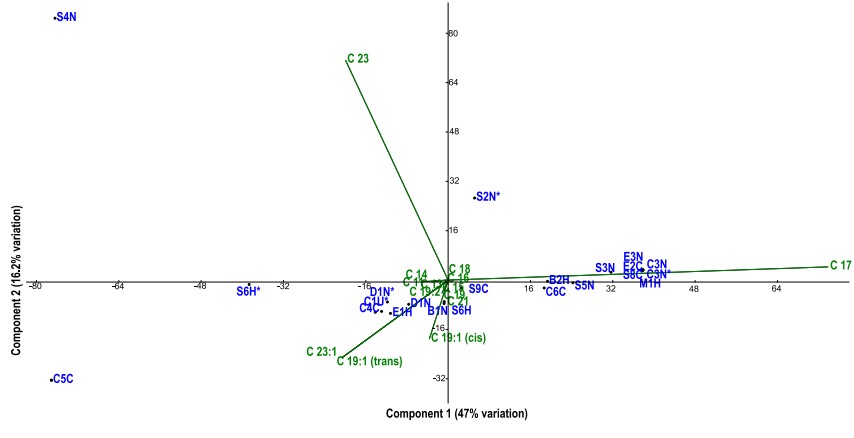

**Figure 3** **PCA biplots A, B and C.** Principal component analysis (PCA) biplots of fatty acid methyl esters from microalgal strains grown under photoautotrophic (A) mixotrophic (acetate) (B) and mixotrophic (glucose) (C) conditions. Fatty acids are represented by green lables and algal strains are indicated in blue by their strain identification. Strains with an asterisk (*) denote axenic status.

cluster to the left of the $y$-axis, indicating their profiles were different than many non-axenic strains. Although some clustering of strains occurred, most strains were fairly unique in their fatty acid profiles, either being dominated by C 17 or C 23 fatty acids. When strains were grown under mixotrophic conditions with acetate (Fig. 3B), the number of fatty acids detected were somewhat comparable to photoautotrophic conditions (Fig. 3A), but with a few fatty acid substitutions including the addition of arachidic acid and the absence of 12-methyltetradecanoic acid and 13-methyltetradecanoic acid. More strains had detectable fatty acids under mixotrophy with acetate, particularly non-axenic strains. Growth with acetate clearly changed the fatty acid composition of strains compared to photoautotrophic growth. When the strains were grown with glucose, a comparatively larger array of fatty acids were detected, but in low concentrations (Fig. 3C). More strains had detectable fatty acids than the photoautotrophy treatment, and more strains clustered together indicating a high degree of similarity. The strains S4N and C5C were found to have very distinct fatty acid profiles based on their location on the biplot. Table 2 provides a summary of the most important fatty acids driving fatty acid profiles under the different growth conditions. As visualized in the biplots, linoleic, oleic, and palmitic acids were dominant fatty acids under all growth conditions (Table 2). Under mixotrophic growth with glucose, the number of important fatty acids explaining the variation in fatty acid profiles doubles in comparison to photoautotrophic growth and mixotrophic growth with acetate.

The total neutral lipids for matched axenic and non-axenic algal strains grown under photoautotrophic and mixotrophic conditions are presented in Table 3. Overall, it is apparent that photoautotrophy favours the production of total neutral lipids, although strain B1N was the exception where mixotrophy on glucose produced significantly more neutral lipids. Notable differences between axenic and non-axenic strains were found for all growth conditions. Axenic strains C3N* and S7H* had approximately an order of magnitude higher total neutral lipids than other algal strains. Strain C3N in particular was a high neutral-lipid producer compared to all of the strains, regardless of axenic status.

## DISCUSSION

The chlorophyte genera isolated from municipal wastewater treatment plants in this study: *Chlorella, Botrydiopsis, Dictyochloris, Ellipsoidon* and *Scenedesmus* reflect a variety of algal strains that are viable in wastewater, but also easy to grow under laboratory conditions. Although found to be among the fastest growing strains, the cyanobacterium *Microcystis* was not a notable producer of fatty acids in this study. *Chlorella* strains were prevalent across the three wastewater treatment systems. This may be due in part to *Chlorella*'s high tolerance of low oxygen conditions, such as in wastewater treatment systems (*Shanthala, Hosmani & Hosetti, 2009*). *Park et al. (2012)* commented that both *Scenedesmus* and *Chlorella* are among the most commonly isolated species from wastewater treatment effluent, which explains their high prevalence in all sampling locations in this study.

Mixotrophic growth, with either glucose or acetate, resulted in higher growth rates compared to photoautotrophic growth for the large majority of strains. This was not unexpected as others have demonstrated higher growth yields for algae under mixotrophic

**Table 2  Regressions using PCA scores.** Regression coefficients (R²) from least-squares linear regression of principle component axis scores and fatty acid proportions for photoautotrophic and mixotrophic conditions. Only statistically significant coefficients are reported ($p \leq 0.05$).

| Fatty acids | PC Axis 1 ($R^2$) | PC Axis 2 ($R^2$) |
|---|---|---|
| **Photoautotrophy** | | |
| linoleic acid methyl ester | 0.8 | |
| oleic acid methyl ester (cis) | 0.7 | |
| palmitic acid methyl ester | 0.92 | |
| 12-methyltetradecanoic acid methyl ester | | 0.72 |
| erucic acid methyl ester | | 0.86 |
| **Mixotrophy-Acetate** | | |
| linoleic acid methyl ester | 0.76 | |
| oleic acid methyl ester (cis) | 0.87 | |
| palmitic acid methyl ester | 0.81 | |
| arachidic acid methyl ester | | 0.76 |
| 14-methylpentadecanoic acid methyl ester | | 0.74 |
| **Mixotrophy-Glucose** | | |
| arachidic acid methyl ester | 0.94 | |
| linoleic acid methyl ester | 0.88 | |
| myristic acid methyl ester | 0.96 | |
| oleic acid methyl ester (cis) | 0.95 | |
| palmitic acid methyl ester | 0.98 | |
| palmitoleic acid methyl ester | 0.94 | |
| stearic acid methyl ester | 0.96 | |
| 13-methyltetradecanoic acid methyl ester | | 0.94 |
| 14-methylpentadecanoic acid methyl ester | | 0.93 |
| 15-methylhexadecanoic acid methyl ester | | 0.94 |
| 9,10-methylene-hexadecanoic acid ME | | 0.94 |
| margaric acid methyl ester | | 0.94 |
| pentadecanoic acid methyl ester | | 0.94 |

growth conditions (*Kirkwood, Nalewajko & Fulthorpe, 2003*; *Das, Aziz & Obbard, 2011*; *Yan et al., 2012*). *Yan et al. (2012)* also demonstrated that the addition of either glucose or acetate actually increased the energy conversion efficiencies over photoautrophic growth. In most cases under heterotrophic growth, the initial algal inoculation either resulted in cell death or showed minimal to no growth. *Price, Yin & Harrison (1998)* found that many microalgal taxa derived from naturally occurring water bodies have generally adapted their cellular processes to daily light fluctuations and therefore are unable to grow strictly heterotrophically. As such, mixotrophy rather than photoautotrophy or heterotrophy appears to be the ideal strategy for maximizing growth rate and biomass yield. Since municipal wastewater and some industrial wastewaters (e.g., pulp and paper effluent) are replete with organic compounds, including sugars, amino acids, and other degradation products, it would be an ideal medium for supplementing algal growth.

**Table 3  Neutral lipids in isolates under different trophic conditions.** Comparison of mean ($n = 3$) total neutral lipids among select axenic and non-axenic strains (axenic strains denoted by * beside the strain name). Standard deviations are in brackets. All treatments for each strain were compared using ANOVA ($\alpha = 0.05$) and the Holm-Sidak method. Statistically significant differences are denoted by: A, photoautotrophic is significantly different from mixotrophic (glucose); B, photoautotrophic is significantly different from mixotrophic (acetate); C, mixotrophic (glucose) is significantly different from mixotrophic (acetate); Z, all treatments are significantly different from each other.

| Strain ID | Neutral Lipids (ng·cell⁻¹) | | | |
| | Photoautotrophic | Mixotrophic (Glucose) | Mixotrophic (Acetate) | Multiple comparisons statistical differences |
|---|---|---|---|---|
| B1N* | 22.1 (0.56) | 45.9 (0.256) | 14.9 (0.186) | Z |
| B1N | 7.49 (0.334) | 62.6 (0.359) | 5.06 (0.012) | Z |
| C3N* | 776 (24.9) | 565 (15.88) | 714 (16.1) | Z |
| C3N | 318 (18.2) | 262 (4.238) | 172 (2.18) | Z |
| D1N* | 25.6 (0.396) | 5.22 (0.046) | 4.84 (0.07) | AB |
| D1N | 34.2 (2.09) | 5.75 (0.060) | 5.59 (0.181) | AB |
| S2N* | 59.6 (2.16) | 30.5 (0.293) | 26.4 (0.451) | AB |
| S2N | 45.6 (2.35) | 44.4 (0.269) | 34.5 (0.913) | BC |
| S3N* | 49.1 (0.41) | 36.4 (1.315) | 12.6 (0.154) | Z |
| S3N | 70.5 (3.19) | 43.8 (0.724) | 18.9 (0.491) | Z |
| S5N* | 64.5 (3.22) | 40.8 (0.647) | 15.8 (0.233) | Z |
| S5N | 36.5 (2.31) | 34.6 (0.856) | 14.9 (0.439) | BC |
| S7H* | 629 (364) | 154 (2.196) | 76.3 (1.67) | Z |
| S7H | 66.2 (2.72) | 53.3 (0.501) | 51.3 (0.790) | AB |

Similar to other studies (e.g., *Nascimento et al., 2013*), the fastest growing strains were not necessarily the best lipid producers (compare Fig. 1 and Table 3). Also, the relative amount of fatty acids produced in each strain varied by axenic status and growth condition and in general, the highest amount of biofuel-targeted fatty acids, such as oleic and linoleic FAMEs, were produced by non-axenic algal strains, particularly slower growing *Scenedesmus* isolates grown mixotrophically with glucose (Fig. 3C). Under photoautotrophic conditions, any bacterial effect on fatty acid profiles of non-axenic microalgal strains were not discernable. However, the effect became evident under mixotrophic growth, particularly with glucose. A possible explanation is that the bacteria consumed not only the organic compounds in the microalgal cultures, but inorganic nutrients as well. This would create nutrient-deprived conditions for the host microalgal strain, which in turn could induce lipid accumulation as a stress response.

Increased lipid production in algae is a common response to stressful or rapidly changing conditions (*Saha et al., 2013*). Since the relative amount and type of fatty acids increased most notably in non-axenic strains of microalgae in our study, this may be the result of a nutrient-stress response via competition with bacteria. *Liu et al. (2012)* discovered that when glucose was added to growth media, it stimulated bacterial growth in algal cultures, which in turn reduced the availability of inorganic nutrients to the algae. A similar result was found when *Scenedesmus obliquus* was grown with a natural bacterial community, where

both phosphorus and nitrogen became limited at the plateau of the alga's growth cycle (*Daufresne et al., 2008*). When grown in co-culture, bacteria can also produce allelopathic exudates that either inhibit algal growth or lyse cells (*Mayali & Azam, 2004*). However, it is unclear if bacterial allelopathy can also induce hyperaccumulation of lipids in algae.

Although for the majority of strains, the total amount of neutral lipids was comparable between axenic and non-axenic strains, there were two striking exceptions (Table 3). Strains C3N and S7H accumulated more neutral lipids particularly under axenic and photoautotrophic conditions. If hyperaccumulation of lipids tends to occur under stressful growth conditions, the results for these two strains may indicate that photoautotrophic growth without bacteria is a stressful growth condition. Of course, this is a preliminary finding, but does highlight the unique differences that can exist among algal taxa isolated from the same environment. Regardless of mechanism, these findings also highlight that, depending on the strain you are working with, the presence of bacteria can have significant impacts on not only the type of fatty acids being produced, but the total amount.

Comparing the microalgal strains under the three different growth conditions showed that axenic strains had a small but detectable difference in fatty acid composition compared to their non-axenic counterparts. A similar result was noted with *C. vulgaris* and *C. sorokiniana* when grown with the bacterium *A. brasilene*, which caused the variation in fatty acids to change from five to eight different fatty acids with increasing amounts of unsaturation (*De Bashan et al., 2002*). Adding glucose to media can change the lipid composition in photoautotrophically grown microalgae by significantly increasing oleic acid concentration (*Sunja et al., 2011*). This was also found in our experiments, where an increased number of strains produced oleic acid under mixotrophic growth with glucose compared to photoautotrophic growth. Additionally, *Cherisilp & Torpee (2012)* have confirmed that microalgae grown under mixotrophic conditions using glucose showed an overall increase in lipid content over photoautotrophic growth and heterotrophic growth, which we also found.

Overall, our assessment of wastewater microalgae has not only shown that mixotrophy can significantly increase growth rate, but that most wastewater *Chorella* and *Scenedesmus* strains exceed the mixotrophic growth-rates of representative culture collection strains. This infers that wastewater strains may be ideal candidates for growth in organic-substrate rich wastewater for biomass and/or biofuel feedstock production. Most of the *Chlorella* wastewater strains were the fastest growers, particularly under mixotrophic and heterotrophic growth conditions. This may increase their potential as biofuel feedstock due to their growth-condition versatility. Additionally, the *Chlorella* wastewater isolates from this study produced total-lipid concentrations on par with wastewater-grown strains (as total lipid in culture-medium) in a study by *Woertz et al. (2009)*.

Although fast growth is an ideal characteristic for any algal strain used in biofuel feedstock production, it must be coupled with comparatively high lipid yields. As previously discussed, the fastest growing microalgal strains were typically not the highest lipid producers. Thus acceptable trade-offs in growth rate vs. lipid yield must be established. To date, the majority of research in algal biofuels involves the maintenance of axenic or low bacterial contamination in microalgal cultures. Yet, our study has demonstrated that the fatty acid

amount and composition can change with bacterial presence. Additionally, the even blend of saturated and unsaturated fatty acids tended to be found in non-axenic microalgal cultures. These findings bode well for growing microalgae in bacteria-laden wastewater. However, it remains to be determined if microalgal growth in wastewater with bacteria results in higher lipid yields and desirable fatty acid composition. As such, the next phase of our research is focusing on the efficacy of growing microalgal wastewater-isolates in municipal wastewater, and assessing the conditions under which growth and lipid production are optimized.

## ACKNOWLEDGEMENTS

The authors wish to thank Heloise Breton and Jenna Comstock for assistance with the FAME protocol and GC-MS calibration and set-up.

### Funding

Funding for this work was provided by FedDev-Ontario and BioFuelNet grants to A. Kirkwood. The funders had no role in study design, data collection and analysis, decision to publish, or preparation of the manuscript.

### Grant Disclosures

The following grant information was disclosed by the authors:
FedDev-Ontario and BioFuelNet grants to A. Kirkwood.

### Competing Interests

The authors declare there are no competing interests.

### Author Contributions

- Kevin Stemmler and Rebecca Massimi conceived and designed the experiments, performed the experiments, analyzed the data, wrote the paper, prepared figures and/or tables, reviewed drafts of the paper.
- Andrea E. Kirkwood conceived and designed the experiments, analyzed the data, contributed reagents/materials/analysis tools, wrote the paper, prepared figures and/or tables, reviewed drafts of the paper.

### Data Availability

   Data sets for this research have been included as Supplemental Information.

### Supplemental Information

Supplemental information for this article can be found online at http://dx.doi.org/10.7717/peerj.1780#supplemental-information.

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
