# Peer review of "Growth and fatty acid characterization of microalgae isolated from municipal waste-treatment systems and the potential role of algal-associated bacteria in feedstock production"

_PeerJ, doi:10.7717/peerj.1780_

## Round 0.1 · original submission · Major Revisions

Please address the reviewer comments. The clearness of the figures needs to be improved, and the results should be analysed in a holistic manner as Reviewers 2 and 3 commented.

Reviewer 1 ·

Basic reporting

This is a well written manuscript with a lot of fundamental research on the microalgae growth. Many samples were collected from three wastewater treatment plants for cultivation, extraction and analysis. Some comments for the authors to address before final acceptance (minor revision):
1. Abstract: address the wastewater treatment systems (e.g. secondary sedimentation plant, etc.)
2. Introduction: Use past or present tense (avoid mixture). See more specifically lines 72-83.
3. Discussion: Authors should discuss the potential application of their findings. For instance, the growth/FAME experiments were conducted with acetate and glucose. What would be the influence of real wastewater (domestic or industrial)? How continues flow studies would affect the growth, lipid production, etc.

Experimental design

No comments

Validity of the findings

No comments

Additional comments

No comments

Reviewer 2 ·

Basic reporting

Many markers in Figures 3-5 are crowded and overlaped, which turns these images vague and meaningless. Is there any other statistic tool available in stead of PCA plots?

Experimental design

1. Abastract should be more concise. The first three sentences did not point out the core objective of this study. Please revise it.
2. The authors have screened several algal strains from a local wastewater treatment plant. The composition of the wastewater should be added. I guess that may give us more clues to find out more useful algea.
3. Page 6 Lines 94-96: This preliminary isolation was conducted with light. Can this be the reason that "extremely low biomass yields for all algal strains grown under heterotrophic conditions"(page 8 line 158)?

Validity of the findings

1. Figure 1: are these growth rate values the best one for each strain? Otherwise I'm afraid that it is hard to announce which one grows faster.
2. Pages 14-15 Lines 293-305: this paragraph appears redundant and irrelavent to the core sense of this study. Please delete it.
3. The authors emphasized that mang previous studies reported the similar phenomena. It's OK but I think that weakens the significance of this study, looking like a repeat of previous findings.
4. Many spectulations are not evidenced. As mentioned in the paper, maybe it's just the first phase study. The paper could be more convincable with more work on these wastewater algea.

Additional comments

Page 4 Lines 58-60
The statement is confusing thus should be rewritten. It generalizes the process applicability to all microalgae while it is only talking about one case study.
Page 5 Line 76
The verb “have ” is unnecessary after “as well as”
Page 5 Lines 87-89
The statement here should be rewritten to make a precision on the sampling sites. In its end it seems there are more than three sampling sites
Page 8 Line 143
There is a typing error in this line
Page 8 Lines 152-153
There is a typing error in the statement
Page 8 Line 155
The abbreviation should be defined before its use
Page 9 Lines 169-170
Rewrite the statement clarifying that it is about front and middle injectors’ temperatures
Page 9 Lines 181-183
The statements should be rewritten in other words instead of using them as they are in the above section
Page 11 Lines 212-213
The statement here should be rewritten to point out the coverage of the present study otherwise it is confusing
Page 11 Line 216-218
The end of the statement is about wastewater treatment plants instead of wastewaters therefore it should be rewritten in a correct way
Page 17 Lines 350-353
The statement here should be rewritten to clarify its meaning
Page 19 Lines 392-394
The bibliography format is incorrect
Page 20 Lines 400-401
The bibliography format is incorrect and the article title is incorrectly written
Page 21 Lines 427-429
The article title is incorrectly written
Page 22 Lines 456-457
The bibliography format is incorrect
Page 23 Line 464
There is an error in pages numbers in the bibliography

Reviewer 3 ·

Basic reporting

The MS was well written. But figure organization was not so good.

Experimental design

Experimental procedure was well descibed.

Validity of the findings

Suggest to compare the experimental results with other related published papers.

Additional comments

This MS isolated 19 microalgae isolates from wastewater, and found that Chlorella and Scenedesmus isolates produced the highest lipid under photoautotrophic and mixotrophic growth conditions with glucose as carbon substrate. Some interestingly, this MS reported a notable effect of commensal bacteria on fatty acid composition from microalgae. This study will be useful in developing algae-based biofuel production, and the MS was well written. However, some revisions were needed prior to consideration of publication in Peer J.

Some specific comments were listed as following:

1, This MS did isolation experiments and did successfully get microalgae isolates from wastewater. However, whether these microalgae had some advantages for fatty acid or biofuel production compared to other published strains was unknown. Suggest to make a comparison in detail.

2, The main purpose of this MS was to reduce cost and improve environmental sustainability through an ideal microalgal feedstock system. But, this MS only investigated the growth and physiological properties of microalgae isolates with glucose/acetate as carbon. It was unknown whether these isolates had good performance under growth in real wastewater conditions.

3, What were effects of temperature and pH (or other related environmental conditions) on strain growth and fatty acid production?

4, In Keyword, better to use the full name of FAME.

---

## Round 0.2 · accepted · Accept

The reviewers' comments have been reasonably addressed.